# YRC-Bench: A Benchmark for Learning to Coordinate with Experts

**Mohamad H. Danesh**\*  *mohamad.danesh@mail.mcgill.ca*
*McGill University*
*Mila – Quebec AI Institute*

**Nguyen X. Khanh**
*University of California, Berkeley*

**Tu Trinh**
*University of California, Berkeley*

**Benjamin Plaut**
*University of California, Berkeley*

**Reviewed on OpenReview:** https://openreview.net/forum?id=YOE0TRK8oU

## Abstract

When deployed in the real world, AI agents will inevitably face challenges that exceed their individual capabilities. A critical component of AI safety is an agent's ability to recognize when it is likely to fail in a novel situation and to yield control to a more capable expert system. Leveraging such expert assistance can significantly improve safety and performance in such situations. Since expert assistance is costly, a central challenge is determining when to consult an expert. In this paper, we explore a novel variant of this problem, termed YRC-0, in which an agent must learn to collaborate with an expert in new environments in an *unsupervised* manner–that is, without interacting with the expert during training. This setting motivates the development of low-cost, robust approaches for training expert-leveraging agents. To support research in this area, we introduce YRC-Bench, an open-source benchmark that instantiates YRC-0 across diverse environments. YRC-Bench provides a standardized Gym-like API, simulated experts, an evaluation pipeline, and implementations of popular baselines. Toward tackling YRC-0, we propose a validation strategy and use a proposer-validator decomposition as a diagnostic framework to evaluate a range of learning methods, offering insights that can inform future research.

Codebase: github.com/modanesh/YRC-Bench

## 1 Introduction

Deploying AI agents in real-world environments presents a critical challenge: agents must operate effectively in dynamic and unpredictable settings, where their individual capabilities are often insufficient to ensure success (Amodei et al., 2016; Leike et al., 2017; Zhou et al., 2024). A promising solution is to equip agents with the ability to leverage assistance from more capable (human or AI) experts (Sadigh et al., 2016; Reddy et al., 2018; Nguyen et al., 2021; Ren et al., 2023). While providing expert assistance can be costly—requiring trained personnel or complex, resource-intensive systems—it is often a worthwhile investment, given its potential to prevent catastrophic failures and outperform purely autonomous agents. Moreover, expert costs can be significantly reduced if an agent can intelligently decide *when* it requires assistance. To build such agents, we study the problem of **Y**ield-or-**R**equest **C**ontrol (YRC), where the goal is to train an agent to decide when to

---
\*Work done while interning at UC Berkeley's Center for Human-Compatible AI (CHAI).

consult an expert in order to succeed with minimal assistance. In this work, we deliberately use nearly-ideal expert policies. This methodological choice allows us to isolate the foundational challenge of an agent's ability to assess its own competence under distribution shift, separating it from the orthogonal–and equally complex–challenge of modeling and coordinating with a fallible, non-stationary human partner. In many safety-critical applications, AI systems must operate under strict constraints that prevent online learning during deployment. These constraints motivate our problem setting: an agent must learn to recognize its own incompetence and yield control before deployment, as it will have no opportunity to learn from failures in the field.

The YRC problem have been explored in various contexts. Previous research often assumes expert availability during the training phase (Nguyen et al., 2019; Nguyen & Daumé III, 2019; Thomason et al., 2020; Xie et al., 2022). Yet, in many real-world scenarios, experts are unavailable or prohibitively costly to employ or simulate. Other settings focus on in-distribution evaluations, assessing agents in environments highly similar to those seen during training (Chernova & Veloso, 2009; Natarajan et al., 2024; Hu et al., 2020) Such settings risk encouraging brittle solutions which fail under drastic distribution shifts.

Toward developing sample-efficient and robust YRC methods, we introduce a novel and practical variant of the YRC problem called YRC-0 (Fig. 1). Our problem motivates the search for unsupervised approaches to YRC. Specifically, in this setting, an agent undergoes a fully autonomous training phase in which it learns a set of tasks independently, without any communication with an expert. At test time, the agent faces novel tasks and may request expert assistance to enhance performance.

Developing unsupervised solutions to YRC is essential given the widespread adoption and inherent vulnerability of fully autonomous training paradigms, such as reinforcement learning (RL) and behavior cloning, and the high costs of training and managing domain experts. Even in scenarios where experts are available, unsupervised approaches could serve as an effective pre-training step, significantly reducing the costs of expert queries during subsequent fine-tuning steps.

In this paper, we present *YRC-Bench*, an open-source benchmark that provides comprehensive infrastructure for conducting empirical research on YRC in general, and on YRC-0 in particular. The extensive collection of environments in YRC-Bench supports a wide range of research needs—from running simple experiments to validate theoretical claims to developing large-scale, multi-environment learning and evaluation approaches. In particular, multi-environment evaluation encourages the development of *robust* methods that do not exploit the idiosyncrasies of any single environment. Furthermore, automated evaluation enabled by simulated experts makes experimentation cost-effective, efficient, safe, and reproducible. In developing this benchmark, we not only overcome engineering challenges but also introduce novel solutions to problems in training and evaluating policies. We release an extensible codebase to support future contributions, enabling easy integration of new environments and methods.

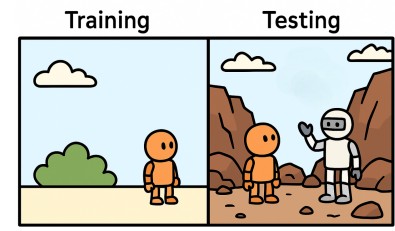

Figure 1: Illustration of the YRC-0 problem. Left: an agent learns tasks on its own (e.g., RL with environment rewards). Right: at test time, it has to perform novel tasks with the help of an expert. While learning in isolation, how can the agent develop a collaboration strategy that will be effective at test time?

Utilizing YRC-Bench, we take initial steps toward developing robust solutions to YRC-0. We conduct a large-scale empirical study, training over 2,600 policies and comparing 10 learning methods across 19 environments. This study expends over 1,300 GPU days (NVIDIA A6000), yielding several noteworthy findings: (1) no method consistently outperforms others, (2) a substantial performance gap remains between current methods and an oracle approach, and (3) this gap stems largely from the simplicity of the policy class considered by these methods, rather than our proposed validation strategy. We distill these findings into concrete recommendations to inform future research.

## 2 Related work

**Human-AI collaboration** has seen growing interest in recent years (Shneiderman, 2022; Wu et al., 2022; Pflanzer et al., 2023; Fragiadakis et al., 2024; Vats et al., 2024). Various settings have been explored in this area. Closest to our work are approaches that enable agents to collaborate with partners who have superior knowledge or skills in a domain (whom we call "experts" in this work). Several works enable this capability to reduce supervision during training (Chernova & Veloso, 2009; Judah et al., 2014). At test time, however, the agent in these settings functions autonomously. Other works focus on building truly collaborative agents which can leverage expert assistance at test time. However, these approaches often assume certain forms of supervision during training time, such as the presence of the expert for interaction (Nguyen & Daumé III, 2019; Nguyen et al., 2019; 2021), an offline dataset recording interactions with experts (Thomason et al., 2020; Padmakumar et al., 2022), or labels of dangerous states (Xie et al., 2022). Our proposed setting takes the challenge to the extreme by assuming no supervision during training, with the goal of developing unsupervised methods that can complement existing supervised approaches. Moreover, it tests the generalizability of the learned policy to novel environments—an aspect overlooked by work embracing the traditional single-environment RL setting (Natarajan et al., 2024; Da Silva & Costa, 2019). Our work differs from online learning settings where an expert is available for queries during training to minimize regret (Cohen et al., 2022), as we focus on the zero-shot case where no expert interaction is possible before deployment.

**RL Generalization and Self-Assessment in OOD Settings.** The core challenge in YRC-0–an agent detecting its own incompetence under an out-of-distribution (OOD) shift–is deeply connected to several established research areas. Our work relates to a broad literature on RL generalization under distribution shifts, which has typically focused on single-agent settings (Danesh & Fern, 2021; Liu et al., 2021; Paudel, 2022; Haider et al., 2023; Yang et al., 2024; Nasvytis et al., 2024). More specifically, YRC-0 intersects with **Safe RL**, where the goal is to identify unsafe states and revert to a safe policy (Xue-Song et al., 2023), and **Uncertainty Quantification**, where methods estimate model confidence to trigger a fallback action (Lockwood & Si, 2022). YRC-Bench provides a benchmark that integrates these challenges into a sequential decision-making problem with cost-sensitive expert queries. It is distinct from prior work on **zero-shot coordination** (Hu et al., 2020), which focuses on coordinating with novel partners but not in novel environments. Finally, YRC-0 serves as a challenging testbed for methods from **meta-learning**, where an agent must learn an adaptation strategy applicable to new tasks (Hospedales et al., 2021).

**Active learning** seeks to minimize annotation cost for training a model. Common methods include uncertainty-based (Lewis, 1995; Settles, 2009), query-by-committee (Roy & McCallum, 2001), submodular maximization (Hoi et al., 2006; Fujii & Kashima, 2016), and deep Bayesian (Gal et al., 2017). While active learning can be cast as an YRC problem, it typically considers a supervised learning problem where data points are identically independently distributed. In contrast, YRC-0 is a sequential decision-making problem with correlated input data. Although sequential variants of active learning have been proposed (Chernova & Veloso, 2009; Judah et al., 2014), the query policy in these settings is learned and used only during training (i.e. no distribution shift for this policy), whereas an YRC-0 policy is primarily used at test time and evaluated under an out-of-distribution (OOD) setting.

## 3 The YRC-0 problem

YRC-0 is concerned with making a sequence of decisions, each of which asks: *in this situation, should an agent make decision on its own or seek advice from an expert?* A reward function is specified to evaluate these decisions. As in any RL problem, the goal is to maximize the expected return (cumulative reward) under a distribution of states and actions induced by the learned policy.

Formally, we define a *task* as a Markov decision process (MDP) with state space $\mathcal{S}$, action space $\mathcal{A}$, reward function $R : \mathcal{S} \times \mathcal{A} \to \mathbb{R}$, initial state distribution $P_0$, and transition function $P : \mathcal{S} \times \mathcal{A} \to \Delta(\mathcal{S})$, where $\Delta(\mathcal{X})$ denotes the probability simplex over a set $\mathcal{X}$. A *task distribution* $\mathcal{E}$ is a distribution over all possible tasks with the same action and state spaces. A *novice* policy $\pi_n : \mathcal{S} \to \Delta(\mathcal{A})$ is trained with tasks sampled from a distribution $\mathcal{E}_{\text{train}}$ and evaluated on tasks sampled from a different distribution $\mathcal{E}_{\text{test}} \neq \mathcal{E}_{\text{train}}$. An

*expert* with policy $\pi_e : \mathcal{S} \rightarrow \Delta(\mathcal{A})$ is present **only at test time** to assist the novice on the test tasks. Importantly, we assume that the novice achieves high performance on $\mathcal{E}_{\text{train}}$ and low performance on $\mathcal{E}_{\text{test}}$, and the expert achieves high performance on $\mathcal{E}_{\text{test}}$. Our setting therefore simulates a typical fully autonomous training procedure, after which the trained agent excels on tasks similar to those it is trained on, but experiences performance degradation when presented with OOD tasks. We assume the novice achieves high performance on $\mathcal{E}_{\text{train}}$ and low performance on $\mathcal{E}_{\text{test}}$ not as a hard requirement, but as the standard consequence of significant distribution shift—the very scenario that makes expert assistance necessary. If the novice performed well on $\mathcal{E}_{\text{test}}$, expert assistance would be unnecessary.

The goal of YRC-0 is to learn a *coordination policy* $\mu : \mathcal{S} \times \Phi_n \rightarrow \Delta(\{n, e\})$ that decides at each time step $t$, whose policy (novice or expert) will be used for deciding the next action. Here, $\Phi_n$ represents the space over internal representations extracted from $\pi_n$ during its decision-making process. Specifically, in state $s_t$, while the expert computes $\pi_n(s_t)$, a representation $\phi(\pi_n, s_t) \in \Phi_n$ is extracted (e.g., the activations of a neural network). The coordination policy $\mu$ then makes a binary decision $x_t \sim \mu(s_t, \phi(\pi_n, s_t)) \in \{n, e\}$. This decision is translated into an environment action $a_t \sim \pi_{x_t}(s_t)$ which is then executed in the environment, generating the next state $s_{t+1}$. Hence, $\mu$ induces an MDP with state space $\mathcal{S}$ and action space $\{n, e\}$. Fig. 2 illustrates this MDP.

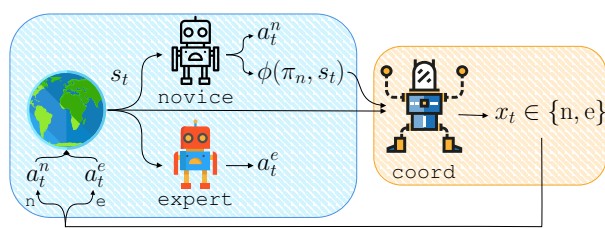

Figure 2: Simulation of the YRC-0 problem. A coordination environment encapsulates two policies: novice and expert. A coordination policy decides which policy will be used to generate the next action. The coordination policy's decision is then translated into an environment action and executed.

With this formulation, we do not assume any specific form of expert advice; it can be demonstration, language utterance, etc. However, to focus on the problem of *when* asks for advice, we assume that the novice's understanding of advice is perfect; each piece of advice has been interpreted into a sequence of executable actions which are stored in the novice's memory. Whenever $x_t = e$, the novice retrieves the next action from memory and executes it.[1] The novice does *not* necessarily bother the expert every time it decides $x_t = e$.

At test time, $\mu$ is evaluated on $\mathcal{E}_{\text{test}}$ where $\pi_e$ is present to assist $\pi_n$. **For the learning of $\mu$, however, $\pi_e$ is unavailable**. The challenge of YRC-0 is to construct a *learning method* that can find an "effective" coordination policy having access to only $\pi_n$ and $\mathcal{E}_{\text{train}}$, where the effectiveness is determined by a reward function described in §4.2. YRC-0 is an OOD generalization challenge, where the distribution shift is caused by two factors: the novel environment dynamics and the introduction of the expert. The latter factor adds a zero-shot coordination challenge to the problem, distinguishing it from RL generalization challenges that focus only on environmental changes Cobbe et al. (2020); Yuan et al. (2023); Pumacay et al. (2024).

It is important to clarify why we forbid the novice or coordination policy from updating parameters during the test phase (online learning). In many high-stakes real-world applications, such as medical diagnosis or autonomous driving, online exploration is prohibited due to safety risks. An agent cannot "try and fail" to learn in a novel environment. It must reliably identify when its pre-trained capabilities are insufficient and yield control immediately. Furthermore, even in settings where online learning is technically feasible, regulatory frameworks often require pre-deployment certification, making online modifications problematic. Finally, regarding the novice's performance, we assume it performs poorly on $\mathcal{E}_{\text{test}}$ not as a hard requirement, but as the standard consequence of significant distribution shift—the very scenario that makes expert assistance necessary. YRC-0 specifically isolates the safety-critical capability of recognizing this competence gap before a failure occurs.

By forbidding interaction with the expert during training, we encourage the development of unsupervised learning methods for YRC. Such methods, even if imperfect, can greatly reduce the cost of expert involvement. From a pragmatic point of view, removing the presence of the expert makes it easy to instantiate the problem

---

[1]One can imagine that the novice stores these actions in a queue, and whenever $x_t = e$, it pops the next action from the queue. If no action is available, it requests a new advice from the expert to refill the queue.

in various environments. In contrast, allowing interactions with experts requires specifying an interaction budget for each environment—a non-trivial task as a single value cannot represent various real-world scenarios.

**The proposer-validation decomposition.**   In this work, we consider a class of learning methods that can be decomposed into two components: a *policy proposer* $\mathcal{P}$ and a *policy validator* $\mathcal{V}$. During training, $\mathcal{P}$ considers a policy class and generates a finite set of candidate policies $\{\mu_1, \mu_2, \ldots\}$ from this class which are then evaluated by $\mathcal{V}$, predicting their test performances. The best candidate is chosen for testing. For example, a deep RL method considers policies parameterized by neural networks. It employs a gradient-based optimizer as the policy proposer, which continuously updates the set of parameters to generate candidates for validation. Another example is to query the expert with probability $p$ at each time step. A grid search over a finite subset of $[0, 1]$ can be used as the policy-proposing approach. We introduce this decomposition primarily as a diagnostic framework for the class of threshold-based and selection-based methods studied in this work. While gradient-based methods (where the "proposer" is the optimizer and the "validator" is the loss function) can theoretically be viewed through this lens, our focus here is on explicit candidate generation and selection. This separation allows us to isolate failure modes: is the method failing because it cannot generate a good policy (proposer failure), or because it cannot identify the good policy among candidates (validator failure)? This distinction is crucial for understanding why current unsupervised baselines struggle.

# 4   *YRC-Bench*: Robust evaluation of learning methods across diverse environments

Prior work on YRC employs diverse training and evaluation settings, making it difficult to compare results. To the best of our knowledge, no study has systematically evaluated these approaches across a large collection of environments. Is there a general approach to YRC that performs well across many domains, or does the no-free-lunch principle apply? We develop YRC-Bench as a research tool to address this question. In addition to assembling a collection of environments, we address practical challenges to enable rigorous yet cost-effective evaluation. Our goal is to enable researchers to easily implement new methods, quickly compare them against existing approaches across diverse domains, and draw generalizable insights into their strengths and limitations.

## 4.1   Overview

YRC-Bench is built on top of three environment suites: MiniGrid (Chevalier-Boisvert et al., 2023), Procgen (Cobbe et al., 2020), and CLIPort (Shridhar et al., 2021), each offering a unique challenge. MiniGrid features grid-based tasks with abstract state representations and partial observability. These environments are highly customizable, making them suitable for simple, controlled experiments to illustrate a phenomenon or validate a theoretical claim. Procgen consists of procedurally generated video games. The main challenges of these tasks are pixel-based observations, stochastic dynamics, and long horizon (some task requires ∼800 steps to complete). CLIPort is a suite of language-guided robotic manipulation tasks that require grounding language instructions in an RGB-D observation space and a continuous action space. A common feature of these environments is that they support the creation of novel environment dynamics (Fig. 3a). Especially, CLIPort requires understanding compositionally novel task instructions. Test performance of standard approaches[2] is far from perfect (Fig. 3b), showing the difficulty of the generalization challenge.

YRC-Bench is designed for extensibility. It supports easy integration of any environment following the gym3 interface.[3] We also defined standard interfaces for the model, policy, and trainer classes to facilitate modular development and ensure compatibility across components.

## 4.2   Resolving evaluation challenges: expert, evaluation metric, and tracking progress

**Simulated expert.** YRC-Bench provides expert policies that emulate real-world experts, enabling large-scale evaluation without the cost, risk, and complexity of employing human operators or heavy AI systems. To

---

[2]For Procgen and CLIPort, we use the approaches proposed by the original work. For Minigrid, we use a 3-layer convolutional max-pooling policy to encode the observation and pass the encoded vector through a GRU RNN.

[3]https://github.com/openai/gym3

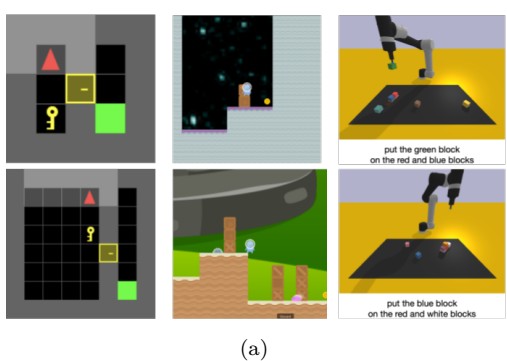 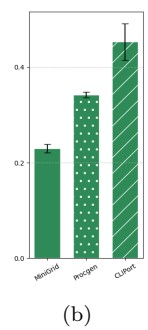 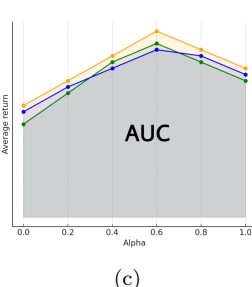

(a)  (b)  (c)

Figure 3: (a) Training (top) and test (bottom) tasks in DoorKey (Minigrid), CoinRun (Procgen), and stack-block-pyramid (CLIPort). (b) The generalization gaps of the novice: its average return on test tasks, normalized by average return of expert. (c) To evaluate policies, we compute the mean and standard deviation of the area under the curve defined by the average return at varying values of $\alpha$ and random seed.

construct these experts, we train PPO (Schulman et al., 2017) policies on $\mathcal{E}_{\text{test}}$ for MiniGrid and Procgen, and employ a rule-based oracle for CLIPort. These experts are nearly ideal: they perform well and respond to all help requests. Nevertheless, our experiments show that current methods struggle even in this simplified setting. This highlights the fundamental difficulty of the problem of identifying risk states. We argue that it is important to first address this core challenge before extending to more complex scenarios, which introduce additional, orthogonal difficulties.

**Reward function.** YRC-0 is fundamentally a multi-objective problem: the coordination policy should maximize the task's return (cumulative reward) while minimizing the cost of expert assistance. To enable direct comparison of methods, it is essential to convert this problem into a single-objective problem. Facing similar issues, previous work employs a hard-constraint approach: either maximizing the reward given a budget of assistance cost (Nguyen et al., 2019), or minimizing expert assistance to achieve a target reward (Ren et al., 2023). In practice, hard constraints may not be intuitive to specify and need to be frequently adjusted as the learning method improves.[4] Alternatively, one can adopt a soft-constraint approach by maximizing a linear combination of return and assistance cost. In this approach, a user needs to specify a weight hyperparameter reflecting a tradeoff between the two quantities. The challenge of this approach is to provide an interpretable formulation that allows users to easily express their preferences through the weight hyperparameter. While many cost formulations are possible, we adopt this approach for its clear interpretability and its ability to automatically adapt to environments with vastly different reward scales, a crucial feature for a diverse benchmark.

We propose a soft-constraint approach where the weight hyperparameter has an interpretable meaning. The approach can be universally applied to any RL environment and expert policy. Let $\bar{G}_e = \mathbb{E}_{\pi_e, \mathcal{E}_{\text{test}}}[\sum_t r_t]$ be the average return of an expert on a test task and $\bar{T}_e$ be the corresponding average episode length. Our idea is to quantify the cost of assistance as a *reduction in return*, and the reduction amount is proportional to the expert return $\bar{G}_e$. Specifically, we define the following reward function

$$r_t(\alpha) = R(s_t, a_t) - \mathbb{1}\{x_t = e\} \cdot \alpha \cdot \frac{\bar{G}_e}{\bar{T}_e} \tag{1}$$

where $R(s_t, a_t)$ is the environment reward obtained for the taken action $a_t$ and $\alpha \in [0, 1]$ is a user-specified hyperparameter. If the expert takes control in $T$ time steps, the total amount deducted from the environment return is $\frac{\alpha T}{\bar{T}_e} \bar{G}_e$. This quantity is adaptive to the specific environment and expert policy. Moreover, the penalty has a clear and interpretable meaning—it corresponds to a fraction of the expert's performance—which facilitates intuitive tuning of $\alpha$. For example, choosing $\alpha = 0.5$ means that if the expert makes decisions

---

[4]If a hard constraint is too easy to satisfy, it hardly affects the optimization of the main objective. When that happens, one needs to enforce a stricter constraint (e.g., requiring the method to achieve a higher performance or lower assistance cost).

in every time step (making $T = \bar{T}_e$), the achieved return is approximately 50% of the expert return. In other words, the return loses 50% of its value due to requesting help.

In practice, users may specify a wide range of $\alpha$ values. Hence, it is crucial to evaluate with multiple values of $\alpha$, simulating diverse scenarios. To summarize performances with multiple $\alpha$ values by a single number, we propose an area-under-the-curve (AUC) metric. As the name suggests, this metric estimates the AUC formed by the points $\{(\alpha_i, \bar{G}(\alpha_i))\}_{i=1}^{K}$ where $\bar{G}(\alpha_i)$ denotes the average return of the evaluated policy for a given $\alpha_i$ (Fig. 3c). We present a bootstrap procedure to compute the mean and error bars of this metric in Alg. 1.

**Tracking progress toward solving a YRC-0 problem.** In ML benchmarks, an "oracle performance" is typically provided to measure progress. However, establishing an oracle performance for a YRC-0 problem is non-trivial. One option is to use the expert's average return, but this performance is unattainable under the assumption that the novice is imperfect on the test tasks. Meanwhile, unlike many ML problems, humans are not oracles in YRC-0. Because the mental states of both the novice and the expert are unobservable to a third-party agent, it is difficult for a human to derive an optimal coordination policy for them. Our solution is to run an RL algorithm at test time to learn a near-optimal coordination policy. This algorithm operates on the MDP induced by the coordination policy and requires access to the expert policy $\pi_e$ and the test environment $\mathcal{E}_{\text{test}}$. We refer to this approach as RLORACLE.

## 5 Policy validation by simulating test conditions

**Policy validation by simulating test conditions.** As mentioned, a major challenge in solving YRC-0 is policy validation, i.e., how to predict the test performance of a policy during training, without access to the expert and test distribution. Without a validation approach, no learning method can be applied to our problem, as the policy selected for testing is ill-defined. To establish our validation approach, let us first define the oracle validator, which evaluates a policy under the exact test conditions $\mathcal{V}^{\star}(\mu) = \text{EVAL}(\mu, \pi_n, \pi_e, \mathcal{E}_{\text{test}})$. Here, the function uses $\mu$ to coordinate $\pi_n$ and $\pi_e$ to perform test tasks sampled from $\mathcal{E}_{\text{test}}$, and returns a score capturing the quality of $\mu$. Our solution constructs a *simulated validator* $\tilde{\mathcal{V}}(\mu) = \text{EVAL}(\mu, \tilde{\pi}_n, \tilde{\pi}_e, \tilde{\mathcal{E}}_{\text{test}})$ that mimics $\mathcal{V}^{\star}$, where $\tilde{\pi}_n$, $\tilde{\pi}_e$, and $\tilde{\mathcal{E}}_{\text{test}}$ are referred to as the simulated novice, expert, and test distribution, respectively.

How should one choose $\tilde{\pi}_n$, $\tilde{\pi}_e$, and $\tilde{\mathcal{E}}_{\text{test}}$ so that $\tilde{\mathcal{V}}$ closely mimics $\mathcal{V}^{\star}$? First of all, we set $\tilde{\mathcal{E}}_{\text{test}} = \mathcal{E}_{\text{train}}$, because we have access to only this distribution during training. Given this choice, we want to construct a simulated expert $\tilde{\pi}_e$ that performs well on this distribution and a simulated novice $\tilde{\pi}_n$ that performs not as effective on it. A natural choice for $\tilde{\pi}_e$ is $\pi_n$, as our setting assumes that the novice performs well under $\mathcal{E}_{\text{train}}$. To construct a poor policy on $\mathcal{E}_{\text{train}}$ as the simulated novice, we learn a *weakened novice* $\pi_n^-$ by running the same algorithm used to train $\pi_n$ but with *limited supervision* (i.e., limiting the number of interactive episodes in RL or demonstrations in behavior cloning). Put all together, our simulated validator is $\tilde{\mathcal{V}}(\mu) = \text{EVAL}(\mu, \tilde{\pi}_n = \pi_n^-, \tilde{\pi}_e = \pi_n, \tilde{\mathcal{E}}_{\text{test}} = \mathcal{E}_{\text{train}})$. We provide a detailed quantitative analysis of this simulated validator's effectiveness in App. D.8. An

Table 1: Uncertainty measures used to construct the coordination policy. Here, $z = (z_1, \cdots, z_{|\mathcal{A}|})$ is the logits computed by the novice in state $s$ (novice is $\pi_n^-$ during training and $\pi_n$ during testing). $p = \text{SOFTMAX}(z)$ and $p^{\downarrow}$ denote the elements of $p$ sorted in descending order.

| Method | Measure $g(s)$ |
|---|---|
| MAXLOGIT | $\max_i z_i$ |
| MAXPROB (Lewis, 1995) | $\max_i p_i$ |
| MARGIN (Scheffer et al., 2001) | $p_1^{\downarrow} - p_2^{\downarrow}$ |
| NEGENTROPY (Settles, 2009) | $\sum_i p_i \ln p_i$ |
| NEGENERGY (Liu et al., 2020) | $\ln \sum_i \exp(z_i)$ |
| Deep SVDD (Ruff et al., 2018) | neural network |

alternative approach to simulating distribution shift would be to hold out a subset of $\mathcal{E}_{\text{train}}$ to serve as a validation set. However, we choose to use the full $\mathcal{E}_{\text{train}}$ for the primary novice to maximize its capabilities. Instead, we use "limited supervision" (reduced training time/data) to create $\pi_n^-$. This serves as a practical heuristic: we model the performance drop caused by distribution shift (in the real setting) using the performance gap caused by under-training (in the simulated setting).

Let $\bar{G}(\pi, \mathcal{E})$ be the mean episode return of a policy $\pi$ on tasks sampled from a distribution $\mathcal{E}$. To achieve a faithful simulation of the test conditions, we want the ratio $G(\pi_n^-, \mathcal{E}_{\text{train}})/\bar{G}(\pi_n, \mathcal{E}_{\text{test}})$ to be close to 1, i.e., the performance of the simulated expert on train tasks is close to that of the real novice on test tasks.[5] Empirically, we already obtain good results if this ratio is not greater than 5.

We select Deep SVDD (Ruff et al., 2018) as a representative baseline for OOD detection-based approaches. Unlike generative approaches that model the full input distribution (which can be computationally expensive or brittle in high-dimensional RL observation spaces), Deep SVDD focuses specifically on learning a compact hypersphere boundary around feature embeddings of the training data. This makes it particularly suitable for detecting deviations in the novice's internal latent representations.

**Policy proposal approaches.** We employ methods that compute a decision function $f_{g,\tau}(s) = \mathbb{1}\{g(s) \geq \tau\}$ for each state $s$, where $g$ is uncertainty measure and $\tau$ is a threshold. If $f(s) = 1$, the novice follows the expert policy $(x_t = e)$; otherwise, it follows its own policy $(x_t = n)$. We consider various choices for the uncertainty measure (shown in Table 1), inspired by active learning and OOD detection methods. For each method, we generate a set of candidates thresholds $\mathcal{T} = \{\tau_1, \tau_2, \cdots\}$. Let $\mu_\tau$ be the coordination policy induced by $f_{g,\tau}$. We select the threshold $\tau^\star = \operatorname{argmax}_{\tau \in \mathcal{T}} \tilde{\mathcal{V}}(\mu_\tau)$ which maximizes the performance computed by the simulated validator $\tilde{\mathcal{V}}$ proposed in the previous section. In this approach, generating the candidate set $\mathcal{T}$ presents a challenge, as the ranges of some uncertainty measures are not fixed. For example, NEGENTROPY outputs a value in $[-\ln|\mathcal{A}|, 0]$, where $\mathcal{A}$ is the action space of an MDP. This range varies across environments, making a standard grid search difficult to conduct. We propose an adaptive method to address this problem. Using the simulated novice $\tilde{\pi}_n$, we generate $K$ task episodes, where the tasks are sampled from the training distribution. We gather a pool of states from these episodes and pass them through the uncertainty measure to generate a pool of scores. We then use the $(n \cdot 10)$-th percentile of these scores as the candidates, where $n = 0, 1, \cdots, 10$.

## 6 Experiments

YRC-Bench presents a unique opportunity to systematically analyze and compare the behavior of diverse methods across a broad spectrum of tasks, yielding more robust insights than evaluations restricted to narrow domains. In this work, we conduct a large-scale study of ten YRC-0 approaches across 19 environments drawn from YRC-Bench. Our experiments are computationally intensive, requiring a total of 1,347 NVIDIA A6000 GPU hours. The detailed runtime for each method and environment suite is provided in §D.6.

### 6.1 Setup

We implement the policy proposal and validation framework described in §5, combined with the uncertainty measures listed in Table 1, resulting in six distinct learning methods. In addition, we evaluate four rule-based baselines: ALWAYSEXPERT, which always yields control to the expert; ALWAYSNOVICE, which always retains control; ALWAYSRANDOM0.5, which flips a fair coin at each step to decide whether to yield control; and RANDOM, which queries the expert with a fixed probability $p \in [0, 1]$. The first three baselines do not require a validator. For RANDOM, the optimal value of $p$ is selected using the simulated validation procedure described in §5. Finally, we include the RLORACLE approach (§4.2) as a reference for oracle performance.

We select Deep SVDD Ruff et al. (2018) as a representative baseline for OOD detection-based approaches. Unlike generative approaches that model the full input distribution (which can be computationally expensive or brittle in high-dimensional RL observation spaces), Deep SVDD focuses specifically on learning a compact hypersphere boundary around feature embeddings of the training data. This makes it particularly suitable for detecting deviations in the novice's internal latent representations. We acknowledge that other uncertainty quantification methods exist, including ensemble-based approaches (Deep Ensembles) and Bayesian methods (MC Dropout), but these require either training multiple models or architectural modifications that may not be feasible when working with pre-trained novice policies. Deep SVDD represents a practical baseline that can be applied post-hoc to existing policies.

---

[5]Note that we are assuming knowledge of $\bar{G}(\pi_n, \mathcal{E}_{\text{test}})$. This is a minimal and reasonable assumption, as without any knowledge about the test conditions, predicting the test performance would be impossible.

For both deep SVDD and RLORACLE, we attempt various types of input features. We try every possible (non-empty) combination of the raw environment observation (`obs`), the hidden features computed by the novice policy (to account for the novice's uncertainty) (`hidden`), and the novice's softmax action distribution (`dist`).

For each environment, we evaluate each method on 1,600 test tasks and run 1,000 bootstrap simulations to compute the mean and standard deviation of the evaluation metric (AUC; Fig. 3c).

### 6.2 Results

Owing to space limitations, we report the principal findings in this section and provide additional analyses in §D. We note that the value of these findings lies not in their quantity, but in their *rigor*, as each is grounded in extensive experimental evidence.

**Finding 1: There is no single best method**. Fig. 4 presents an overview comparison of methods, showing the number of environments in which each achieves the highest mean AUC. Our analysis reveals a lack of consistency across methods. Even the most successful ones achieve top performance in only 3 out of 19 environments. This result underscores the importance of a thorough empirical evaluation when selecting a solution approach for a specific YRC problem. It also suggests the necessity of having a comprehensive benchmark like YRC-Bench, which supports quick evaluation of diverse methods by providing a unified interface, standardized evaluation pipeline, and off-the-shelf baseline implementations.

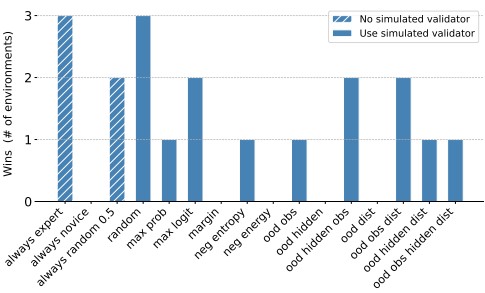

Figure 4: Number of environments where a method attains the highest mean AUC; solid bars denote use of our validation method.

**Finding 2: Our simulated validation approach is effective.** Methods leveraging this approach collectively outperform their counterparts in 14 out of 19 environments. Furthermore, 3 out of the 4 most successful methods employ the simulated validator.

**Finding 3: Existing Uncertainty-Based Methods Are Brittle, Often Failing to Outperform a Random Baseline.** Our analysis reveals a stark indictment of the brittleness of current uncertainty-based methods when faced with distribution shifts. Sophisticated approaches like deep SVDD, which learn heuristics from the training distribution, often fail catastrophically, leading them to make confidently incorrect decisions about when to request help. This failure is so significant that in multiple environments, these methods are outperformed by a simple RANDOM baseline that requests help by flipping a coin. Rather than being a flaw in the benchmark, this result underscores the severity of the OOD self-assessment challenge: methods designed for uncertainty estimation can perform worse than random chance because they learn systematic biases that do not generalize. The RANDOM approach, in its simplicity, avoids these biases. This poor performance across the board, highlighted by the comparison to a non-adaptive baseline, stands in stark contrast to the oracle performance shown in Fig. 5, confirming that there remains significant room for improvement, especially in challenging environments like Procgen and CLIPort.

**Finding 4: Improving the policy proposer offers significantly more potential for performance gain than improving the validator** .

To offer more specific guidance for future development, we introduce a systematic diagnostic method based on the proposer-validator decomposition of each algorithm. As a reminder, the policy proposer generates candidate coordination policies, while the validator evaluates these candidates to select the best one. Ideally, we want a policy proposer that identifies the optimal policy as a candidate, and a validator that ranks it above all other policies. When an approach falls short, either the policy proposer, or the validator, or both are deficient.

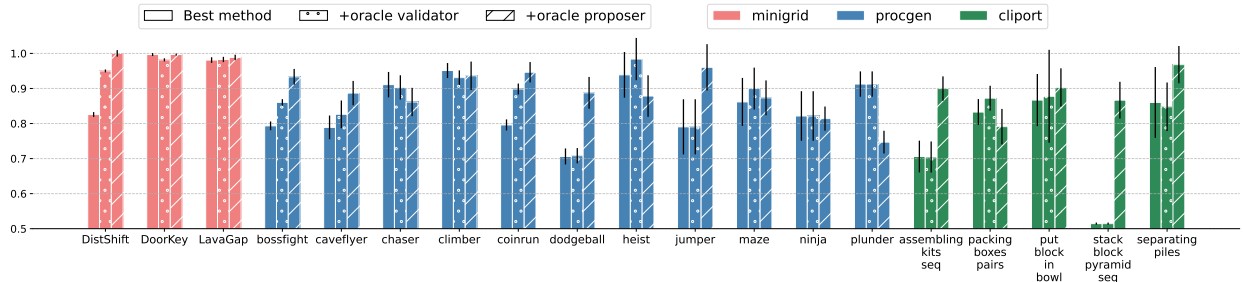

Figure 5: Test performance of learning methods across environments, normalized by the performance of the best RLORACLE method. For each environment, we show three variants: the best performing method with simulated validation (i.e., excluding ALWAYSNOVICE, ALWAYSEXPERT, and ALWAYSRANDOM0.5), the same method with an oracle validator (+oracle validator), and the same method with an oracle proposer, which employs a deep RL approach (+oracle proposer). The gaps between the latter two variants and the original indicate potential performance gains that could be achieved by improving the replaced components. Error bars represent 2× standard deviation.

The proposer-validator decomposition enables us to identify components limiting the performance of an algorithm by replacing each with an *oracle counterpart* and measuring the resulting improvement. A performance boost after replacement indicates that the replaced component is deficient and requires enhancement.

As shown in Fig. 5, replacing the simulated validator with an oracle validator yields minimal improvement in most environments. In contrast, replacing the policy proposer with its oracle counterpart produces substantial performance improvements in 10 out of 19 environments. This stands in stark contrast to the minimal gains observed when replacing the simulated validator with its oracle counterpart (Fig. 5). While the overlapping error bars prevent a definitive conclusion that the validator is not a bottleneck, our results strongly suggest that the policy proposer is the primary limiting component for the methods studied. Therefore, we conclude that improving the policy proposer offers a much larger potential for performance gain than improving the validator.

Our finding suggests that future research should focus on methods capable of exploring richer policy spaces. We discuss a notable exception to this trend in the "plunder" environment in App. D.7.

Taken together, our results reveal a fundamental limitation of current approaches: their search strategy does not identify effective candidate policies, as it operates within an overly restricted policy space. Our finding suggests that future research should focus on methods capable of exploring richer policy spaces.

Drawing from the findings obtained from our experiments, we compose a list of concrete recommendations for practitioners who want to develop or deploy solutions to YRC problems:

1. Explore diverse methods, as good performance reported in one environment may not transfer to another.

2. Do not overlook simple, computationally cheap baseline like RANDOM, which may yield surprisingly good performance.

3. Implement an oracle approach, like RLORACLE, to estimate the remaining room for improvement. Remember that human-based policies are often not oracles in this problem.

4. Replace the policy proposer or policy validator of the current method with an oracle counterpart to pinpoint the method's primary limiting component.

# 7 Conclusion & limitations

In this work, we formalize the learning to Yield and Request Control (YRC) problem, a critical challenge for AI agents operating in dynamic, safety-critical environments. Solving this problem is an important first step toward tackling more complex human-AI collaboration challenges. Our findings highlight significant

potential room for improvement, highlight opportunities for the research community to contribute in this area. Developing robust methods for this problem paves the way for safe, effective human-AI collaborative systems in the future.

While our work proposes a cost-effective and computationally efficient simulation of real-world scenarios, several limitations warrant consideration. First, the simulated experts in YRC-Bench may not fully capture the variability and cognitive biases of human operators. While a full human study was beyond the scope of this foundational work, we consider it a crucial direction for future research to validate our findings with real human participants. Second, although our benchmark incorporates distribution shifts across environments, real-world shifts may involve more complex, multimodal dynamics that are not yet modeled in existing simulations. Third, the cost model assumes fixed query costs, whereas practical deployments often face context-dependent or time-varying costs. By focusing on rule-based (always/random), logit-based, OOD-based and oracle-RL methods, we intentionally cover the most widespread coordination paradigms, extracting clean, generalizable patterns that serve as performance baselines. Although we do not exhaust every possible algorithmic family (e.g., meta-learning, offline RL, human-in-the-loop adaptation), our findings reveal fundamental proposer-validator bottlenecks that will directly inform and accelerate the design of these future approaches. In particular, evaluating stronger baselines from related fields, such as ensemble or Bayesian methods for more robust uncertainty quantification, is an important next step. Finally, our evaluation focuses on episodic tasks, leaving open questions about lifelong coordination in non-stationary settings. We choose PPO-trained and rule-based "experts" to guarantee scalable, deterministic evaluation, sacrificing the idiosyncratic biases of real humans for consistency and speed. This strategic trade-off establishes a stable, low-variance baseline. Our modular framework can seamlessly swap in human-in-the-loop studies or richer learned expert models in future work without altering its core mechanics.

In terms of methodology, we have so far explored only simple policy-proposing and validation approaches. Threshold-based methods perform well with our proposed validator because they operate within restricted policy spaces, which effectively mitigate overfitting. However, as we have shown, such a constrained policy space also limits performance. As one transitions toward methods that operate on richer policy spaces (e.g., deep RL methods), the risk of overfitting increases, and our proposed validator may not be sufficiently reliable to prevent it. We believe that addressing the current limitations of the simulation and developing a more faithful simulated validation approach are promising directions for future research.

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

# Appendix

## Table of Contents

**Pseudocode 1. Coordination environment interface**

```
class CoordEnv(gym.Env):
    def __init__(self, config, base_env, novice, expert):
    def reset(self):   # unlike gym3, actually reset to an initial state of a new task
    def step(self, action):   # like gym3, automatically reset upon end of episode
```

Figure 6: Interface of the coordination policy's MDP class.

**Pseudocode 2. Training a coordination policy**

```
# 1) Parse command-line flags
args = flags.make()
# 2) Configuration can be specified using a YAML file (args.config) or by flags (args)
config = config_utils.load(args.config, flags=args)
# 3) Make training, validation, and test environments
envs = YRC.core.environment.make(config)
# 4) Initialize coordination policy
coord_policy = YRC.core.policy.make(config, envs)
# 5) Create evaluator
evaluator = YRC.core.Evaluator(config)
# 6) Initialize algorithm
algorithm = YRC.core.algorithm.make(config, envs)
# 7) Train coordination policy
algorithm.train(coord_policy, envs, evaluator)
```

Figure 7: Training a coordination policy with YRC-Bench.

## A  YRC-Bench

### A.1  Coordination Environment Wrapper

To standardize coordination policy training and evaluation, we introduce the `CoordEnv` wrapper. This class converts any Gym-compatible environment (Brockman et al., 2016; Towers et al., 2024) into an *MDP for the coordination policy* that preserves the original state space $\mathcal{S}$ but replaces the action space with two choices $\{n, e\}$, representing the coordination policy's decisions of requesting control (novice acts) and yielding control (expert acts). At each timestep, the wrapper resolves the coordination policy $\mu$'s decision into an action in the base environment's action space: $a_t \sim \pi_{x_t}(s_t)$. Subsequently, the next state $s_{t+1}$ is generated following the base environment's transition dynamics: $s_{t+1} \sim P(s_t, a_t)$.

Fig. 6 describes the interface of `CoordEnv`. The class expects a base environment that follows the `gym3` interface except that calling `.reset()` actually resets it to an initial state of a task (the `reset()` method of an `gym3` environment has no effect). This feature is employed during evaluation to ensure that all evaluation runs are conducted with the same set of tasks.

Fig. 7 shows simple steps to train a coordination policy using our codebase. Each step is highly customizable by extending the codebase's core classes.

### A.2  Training Novice and Expert Policies

Each environment requires three policies for training and evaluation: expert $\pi_e$, novice $\pi_n$, weakened novice $\pi_e^-$.

**Expert construction**. For MiniGrid and Procgen, we train $\pi_e$ using PPO on $\mathcal{E}_{\text{test}}$ until convergence (Huang et al., 2024)For CLIPort, we use the rule-based oracles provided by the benchmark.

**Novice construction**. The novice $\pi_n$ is trained exclusively on $\mathcal{E}_{\text{train}}$. MiniGrid and Procgen novices are learned by running PPO on $\mathcal{E}_{\text{train}}$ until convergence CLIPort novices are checkpoints trained with 100 demonstrations.

**Weakened Novice Policy Training**. We create $\pi_n^-$ by deliberately limiting training exposure. For MiniGrid and Procgen, we halve PPO training epochs while maintaining $\mathcal{E}_{\text{train}}$ exposure. CLIPort's $\pi_n^-$ uses checkpoints trained on only 10 demonstrations, reflecting partial task mastery. This mimics test-time performance degradation while preserving training distribution familiarity.

### A.3  Training Coordination Policy

#### A.3.1  Logit-Based Methods

Let $\mathbf{z} = (z_1, z_2, \cdots, z_{|\mathcal{A}|})$ be the logits output by the novice, and $\mathbf{p} = \mathsf{Softmax}(\mathbf{z}) = (p_1, p_2, \cdots, p_{|\mathcal{A}|})$ is the softmax distribution derived from $\mathbf{z}$. A logit-based method outputs a score $u$ that captures the degree of uncertainty of the novice. Whenever the score falls below a pre-specified threshold, the novice yields control to the expert.

The score for each method is defined as follows:

- MaxLogit: $u = \max_i z_i$
- MaxProb: $u = \max_i p_i$
- Margin: $u = \max_i p_i - \max_{j \neq i^\star} p_j$ where $i^\star = \operatorname{argmax}_i p_i$ (the difference between the highest and second highest softmax probabilities)
- NegEntropy: $u = \sum_{i=1}^{|\mathcal{A}|} p_i \ln p_i$
- NegEnergy: $u = \tau \cdot \ln \sum_{i=1}^{|\mathcal{A}|} \exp(z_i/\tau)$. Following (Liu et al., 2020), we set the temperature $\tau = 1$ in all experiments.

**Threshold selection**. To determine the threshold, we conduct a grid search over a set of candidate thresholds, and select the one that maximizes validation performance. To generate these candidates, we rollout $\pi_n^-$ for 64 episodes. In each episode, the policy executes a task sampled from $\mathcal{E}_{\text{train}}$ This generates a pool of uncertainty scores, which each corresponds to the decision of the novice in a state encountered in an episode. We sort the scores and use the $k$-th percentiles as threshold candidates with $k = 0, 10, 20, \cdots, 100$. This method is data-driven and adaptive to the score range, which varies dramatically among methods.

#### A.3.2  Deep SVDD

Our implementation of Deep SVDD is based on PyOD. All hyperparameters for Deep SVDD are set to their default values in PyOD. Similar to Logit-based methods, Deep SVDD computes an uncertainty score; the novice yields control if the score is below a threshold. We first generate executions of 64 $\mathcal{E}_{\text{train}}$ tasks using $\pi_n^-$. The generated data is used to both train the OOD-detection model and to determine the optimal threshold. The threshold is also chosen following a similar procedure as that of logit-based method: we generate a pool of uncertainty scores and use the percentiles as candidates.

#### A.3.3  RLOracle

RLOracle implements Proximal Policy Optimization (PPO) (Schulman et al., 2017). Our implementation largely follows CleanRL (Huang et al., 2022).[6] To obtain our results, we run this algorithm with the coordination environment wrapper corresponding to $\mathcal{E}_{\text{test}}$.

The underlying policy is an Impala model (Espeholt et al., 2018). Depending on the chosen configuration, the input features may include a raw environment observation, a hidden representation extracted from the novice, or the softmax output distribution of the novice.

---

[6]https://github.com/vwxyzjn/cleanrl/blob/master/cleanrl/ppg_procgen.py

---

**Algorithm 1** Bootstrap procedure to compute AUC metric. `AreaUnderCurve` computes the area under the curve formed by the input points. We use $N = 1000, K = 6, M = 1600, m = 256$ in our experiments.

---

1: **Input:** Data points $\{(\alpha_i, \{G_{i,j}\}_{j=1}^M)\}_{i=1}^K$ where $\alpha_i = \frac{i}{K}$ and $G_{i,j}$ is the return of the evaluated policy in the $j$-th episode, during which $\alpha = \alpha_i$.
2: **Output:** Mean estimation and its standard deviation

3: Initialize $E = \emptyset$
4: **for** $N$ simulations **do** ▷ §5
5:     Initialize $D = \emptyset$
6:     **for** $i = 1 \ldots K$ **do**
7:         Draw an $m$-element sample $S_i$ from $\{G_{i,j}\}_{j=1}^M$
8:         Compute $\bar{G}_i = \mathsf{mean}(S_i)$
9:         $D \leftarrow D \cup \{(\alpha_i, \bar{G}_i)\}$
10:     $E \leftarrow E \cup \{\mathsf{AreaUnderCurve}(D)\}$
    **return** $\mathsf{mean}(E), \mathsf{std}(E)$

---

We refer readers to our GitHub repository for detailed training and model hyperparameters.

### A.4 Evaluation

Alg. 1 presents our bootstrap procedure for computing the evaluation metric for each policy.

## B  Policy Feature Extraction

In MiniGrid, both the novice $\pi_n$ and weakened novice $\pi_e^-$ are trained using the GitHub repository: https://github.com/lcswillems/rl-starter-files. To extract the hidden features (`hidden`) and the softmax action distribution (`dist`), we leverage the model's intermediate layers. For the former, we process the raw image observations through convolutional layers and text inputs through an embedding module, fusing these modalities into a unified hidden representation. Then, for the latter, we apply the actor network to this hidden representation to compute unnormalized action scores (logits). Critically, this mirrors the forward pass of the original model but excludes the final step of constructing a probability distribution (via the `Categorical` class), allowing direct access to the pre-softmax logits. This approach ensures alignment with the model's internal decision-making process while enabling targeted analysis of policy behavior.

For Procgen, we employ the same framework. Hidden features (`hidden`) are extracted via the `ImpalaModel` embedder (Espeholt et al., 2018), which processes high-dimensional visual observations through its residual convolutional architecture. Then these features are mapped through the policy network to produce unnormalized action scores. While the model's forward pass applies `log-softmax` normalization and constructs a categorical distribution for policy gradient updates, our analysis directly utilizes the raw outputs from the final policy layer. This preserves the model's original action preference rankings while bypassing probability normalization, a critical design choice that maintains numerical fidelity with the policy's internal decision logic while enabling direct comparison of action selection mechanisms across different policy versions.

For CLIPort, we adapt the framework to accommodate its dual-stream architecture for robotic manipulation. Hidden features are derived through separate attention (`pick`) and transport (`place`) networks that process RGB-D observations and language goals through coordinated ResNet and CLIP-linguistic fusion pathways He et al. (2016); Radford et al. (2021). The attention stream first computes pixel-wise confidence maps for object selection, while the transport stream subsequently predicts placement locations and orientations conditioned on the chosen pick point. Unnormalized action scores emerge as spatial heatmaps encoding both positional and rotational preferences across the workspace. Though the operational pipeline constructs categorical distributions during training through spatial softmax normalization, our analysis directly utilizes the pre-normalization confidence values from both streams. This preserves the geometric relationships in the model's pick-and-place reasoning while maintaining fidelity to the original visuolinguistic feature representations, crucial for interpreting the policy's physical interaction decisions in structured action spaces.

# C   Environments

YRC-Bench is built on top of three open-source benchmarks: MiniGrid (Chevalier-Boisvert et al., 2023) , CLIPort (Shridhar et al., 2021), ProcgenAISC (commit `7821f2c`) (Di Langosco et al., 2022).

## C.1   MiniGrid Environments

We use the following environments:

- `MiniGrid-DistShift-`: `1-v0` for training, and `2-v0` for testing;
- `MiniGrid-DoorKey-`: `5x5-v0` for training, `8x8-v0` for testing;
- `MiniGrid-LavaGap`: `S5-v0` for training, `S7-v0` for testing

All environments use partially observable grids with discrete actions. The test environments feature larger state spaces and more complex trajectory solutions.

## C.2   Procgen

The suite includes 11 distinct platformer games with pixel-based observations and discrete actions:

- `bossfight`: combat-focused game with escalating enemies
- `caveflyer`: navigation through procedural caverns
- `chaser`: avoidance of pursuing enemies
- `climber`: vertical ascension challenge
- `coinrun`: collection-based platformer
- `dodgebal`: projectile avoidance game
- `heist`: stealth-based item retrieval
- `jumper`: precision jumping challenges
- `maze`: complex spatial navigation
- `ninja`: timing-based obstacle course
- `plunder`: resource gathering under threat

We use the *easy* distribution for training and the *hard* distribution for testing. The *hard* distribution introduces stochastic elements and more complex terrains.

## C.3   CLIPort

We experiment with five tasks:

- `Assembling-Kits-Seq`: sequential object placement in kits
- `Packing-Boxes-Pairs`: pbject pairing and containerization
- `Put-Block-in-Bowl`: precise object-in-container placement
- `Stack-Block-Pyramid-Seq`: vertical structure assembly
- `Separating-Piles`: object sorting and segregation

We use the *seen* split for training and the *unseen* split for testing. The *unseen* split (testing) introduces novel object geometries and color combinations not encountered in the seen split. The tasks require 6-DOF control with a continuous action space and spatial reasoning over pixel-based observations.

Figure 8: Per-environment performance of RLORACLE variants. Using observations as input features yields larger performance boost in more complex environments like Procgen and CLIPort.

# D   Detailed Results

## D.1   Performance of RLOracle Methods

We further analyze the performance of individual RLORACLE algorithms across different environments, as shown in Fig. 8. This detailed breakdown reveals that the advantage of raw observation-based policies is more pronounced in the Procgen and CLIPort environments, while it is less evident in the MiniGrid suite. This discrepancy can be attributed to the nature of the environments: Procgen and CLIPort feature visually rich, high-dimensional observation spaces, where direct access to raw observations provides a clear advantage in learning nuanced coordination behaviors. In contrast, MiniGrid offers low-dimensional, symbolic representations, where the distinction between raw observations and the novice's internal features is less significant. In such structured environments, the novice policy's internal representations already capture most of the relevant task information, diminishing the benefit of using raw observations.

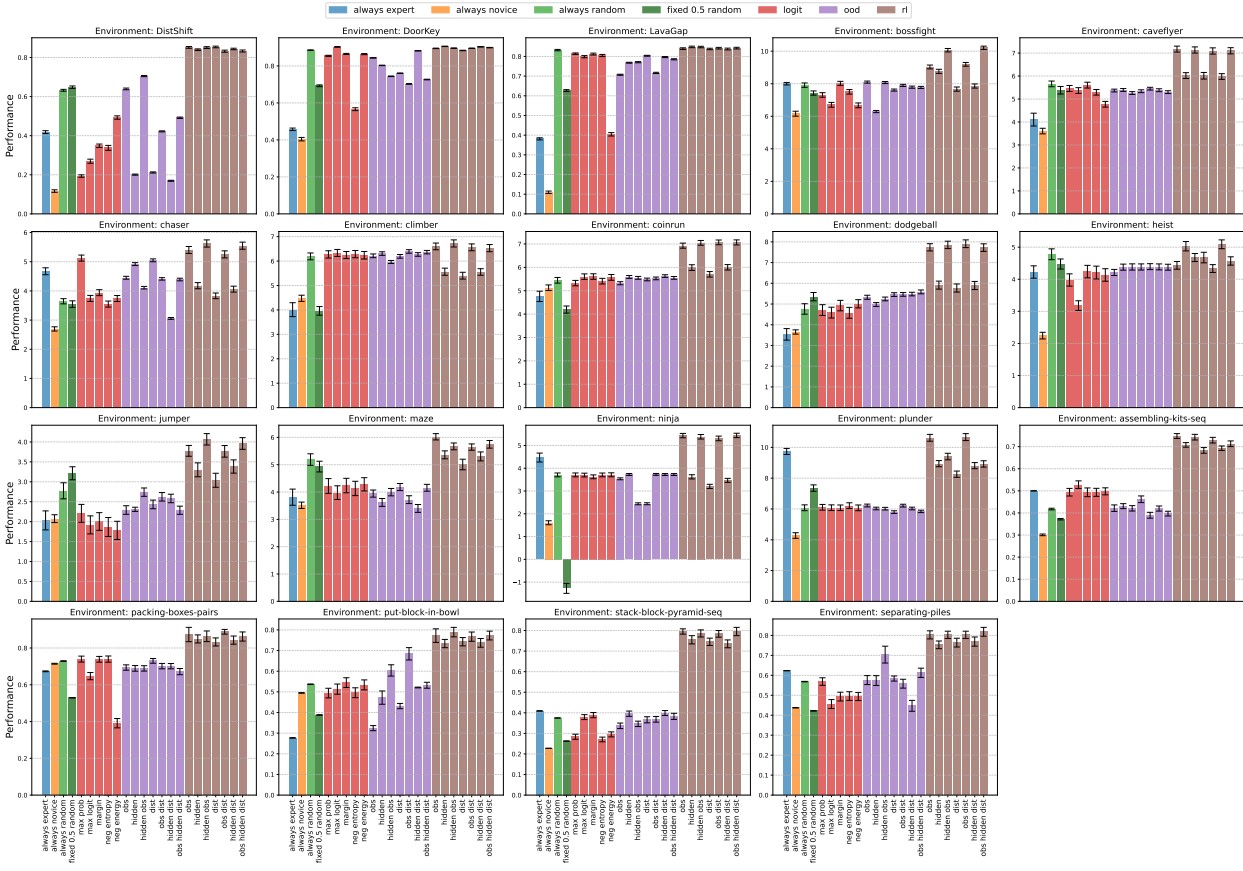

Figure 9: Comprehensive performance comparison across all learning methods and environments. RLORACLE represents the RL-computed upper bounds. logit and OOD methods approach the upper bound in several environments, mostly MiniGrid ones. Gray backgrounds denotes those environments.

## D.2 Near-Optimal Coordination Achievements

Fig. 9 illustrates the overall performance of each algorithm and input feature type across all environments studied in this paper. It reveals an interesting pattern: logit-based and OOD detection-based coordination policies achieve near-skyline performance in 3 environments. We analyze these representative success cases:

**DoorKey (MiniGrid):** The $8 \times 8$ grid environment exhibits deterministic dynamics but requires precise multi-step sequencing (find key, then unlock door, then navigate to goal). The MAXLOGIT policy matches skyline performance matches skyline performance by interfering the novice's potentially flawed decision-making, preventing costly mistakes and ensuring efficient completion of the task.

**LavaGap (MiniGrid):** This environment's lethal consequences (falling into lava) create clean separation between high-confidence navigation actions and uncertainty "cliff edges." The OOD-based method with `hidden-dist` features and Margin logit policy are the closest to skyline performance.

**Climber (Procgen):** Despite procedural generation, the logit-based methods are statistically the same as skyline methods. The policy successfully distinguishes between challenging-but-seen obstacles (handled by novice) and truly novel gap configurations (referred to expert), despite being trained solely on the `easy` distribution.

Our analysis also reveals substantial performance gaps between RLORACLE and other methods. Notably, across all CLIPort manipulation tasks, no method approaches even the worst-performing RLORACLE. For example, in the `packing-boxes-pairs` task, the lowest-performing RLORACLE variant (using only the novice's

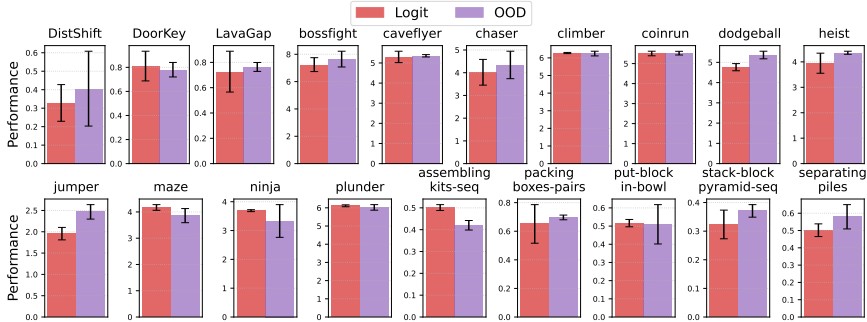

Figure 10: Comparison of logit-based and OOD detection methods. Error bars show the standard deviation across each method's variants.

action probability distribution as input) achieves a performance of 0.83, while the best non-RLORACLE methods (logit-based approaches) reach only 0.73, representing a 13.7% relative performance gap. Other CLIPort tasks exhibit even wider disparities, with RLORACLE outperforming alternatives by at least 30.7% on `assembling-kits-seq`, 35.1% on `put-block-in-bowl`, 40.3% on `stack-block-pyramid-seq`, and 20.9% on `separating-piles`. These substantial gaps highlight fundamental limitations in current coordination strategies for high-dimensional manipulation tasks, underscoring the urgent need for improved policy architectures that better leverage both environmental observations and novice uncertainty signals.

### D.3 Comparison of Logit-based and OOD detection-based Methods

Our experiments reveal an interesting insight in comparing logit-based methods with the Deep SVDD OOD detection approach, as quantified in Fig. 10. Overall, in 1 out of 19 evaluated environments, logit-based methods outperform Deep SVDD. In 2 environments the OOD detection-based method performs better. And in the remaining cases, they tie.

This suggests that practitioners may prefer computationally lightweight logit-based coordination unless operating in domains with known visual-semantic mismatch between observation space and task requirements. Based on our results, we suggest practitioners reconsider the prevailing assumption that complex OOD detection is universally preferable for safety-critical coordination (Yang et al., 2024). We demonstrate that simpler approaches often suffice when distribution shifts primarily affect agent behavior rather than environmental appearance.

### D.4 Best features for RLOracle

. While being an oracle in our setting, RLORACLE is a viable approach in a life-long learning setting, where the novice continuously adapts to test conditions. We investigate the best recipe for this approach to provide helpful insights for researchers who want to tackle this setting.

Our experiments reveal that including raw environment observations as input to the coordination policy consistently improves performance compared to using only its hidden representations or its logit outputs. This trend presents in 15 out of 19 environments (Fig. 11), suggesting that the novice does not acquire helpful, easily extractable uncertainty information if trained only to perform tasks autonomously.

Our results also highlight a relationship between environment complexity and observation-space utility. While raw observations generally provide richer learning signals, their value

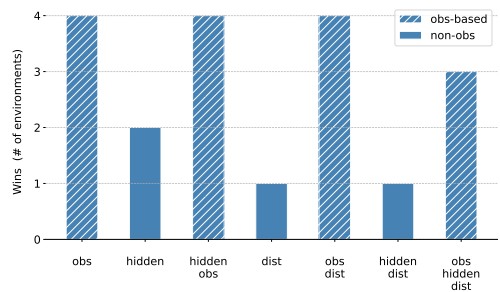

Figure 11: Number of environments in which each variant of RLORACLE achieve the highest AUC mean. Variants that take raw environment observations as input yield superior performance.

diminishes in structured environments with comprehensive feature representations. For instance, in Minigrid environments, the difference between using raw observations and structured feature representations is negligible. However, in more visually complex environments like Procgen or CLIPort (e.g., `CaveFlyer` or `stack-block-pyramid-seq`), raw observations provide crucial information that significantly enhances performance (see App. D.1 for details). We thus suggest practitioners to prefer observation-conditioned coordination policies unless observations are complex to model and hidden representations are sufficiently rich.

### D.5 Normalized Novice-to-Expert Score Ratios

To quantify the gap between novice and expert policies on unseen test tasks, $\mathcal{E}_{\text{test}}$, we compute for each environment $i$ the normalized score ratio

$$r_i = \frac{\bar{W}_i}{\bar{E}_i} \qquad\qquad \delta r_i = r_i\sqrt{\left(\frac{\sigma_{W,i}}{\bar{W}_i}\right)^2 + \left(\frac{\sigma_{E,i}}{\bar{E}_i}\right)^2} \qquad\qquad (2)$$

where $\bar{E}_i \pm \sigma_{E,i}$ and $\bar{W}_i \pm \sigma_{W,i}$ are the expert's and novice's mean returns and standard deviations, respectively. Fig. 12 (left) shows $r_i$ with $2 \times \delta r_i$ error bars for all 19 environments across MiniGrid, Procgen, and CLIPort.

We chose these suites to capture distinct evaluation challenges:

- **MiniGrid:** a simple, highly customizable gridworld for quick proof-of-concept demonstrations on abstract state representations, highlighting decision-making challenges that generalist LLMs often face.
- **Procgen:** long-horizon, procedurally generated visual tasks that stress robust visual control and exploration.
- **CLIPort:** vision-language manipulation benchmarks requiring integration of visual perception with high-level language instructions.

Across individual tasks, novice policies achieve between approximately 12% and 43% of expert returns in MiniGrid (mean $r \approx 0.23$), $17\% - 55\%$ in Procgen (mean $r \approx 0.32$), and $24\% - 74\%$ in CLIPort (mean $r \approx 0.45$). This systematic shortfall highlights the difficulty of generalizing to novel test environments.

We further aggregate these ratios at the suite level by

$$r_S = \frac{1}{|S|}\sum_{i \in S} r_i \qquad\qquad \delta r_S = \frac{\sqrt{\sum_{i \in S}\delta r_i^2}}{|S|} \qquad\qquad (3)$$

where $S$ indexes environments in each suite. Fig. 12 (right) shows the average ratio $r_S$ with $2 \times \delta r_S$ error bars. The consistent deficit across all suites, never exceeding 50% of expert performance, underscores that even our best-trained novices fall significantly short of expert performance when generalizing beyond training.

These patterns serve two purposes: (1) they illustrate that different environment families pose unique generalization challenges, and (2) they motivate our coordination framework, which adaptively requests expert guidance most aggressively in those environments where $r_i$ is both lowest and most variable, thereby allocating expert queries where they yield the highest marginal benefit.

### D.6 Computation Cost and Time

All experiments were executed on a single NVIDIA A6000 GPU with 48 GB VRAM and 100 GB of host memory. To highlight the significant computational cost, Table 2 summarizes the total wall-clock times for the training and evaluation phases, as well as their combined cost.

Table 2 reveals that we expended a staggering total of **over 1347 GPU-days** of continuous run time on a single A6000 card. This corresponds to nearly **3.7 years of dedicated GPU usage**, underscoring the high resource demands of large-scale experiments under distribution shift. In particular, the Procgen suite alone accounted for almost 82% of total compute during training, reflecting the complexity and variability of those environments. Even the simplest MiniGrid tasks required several GPU-days to achieve robust evaluation.

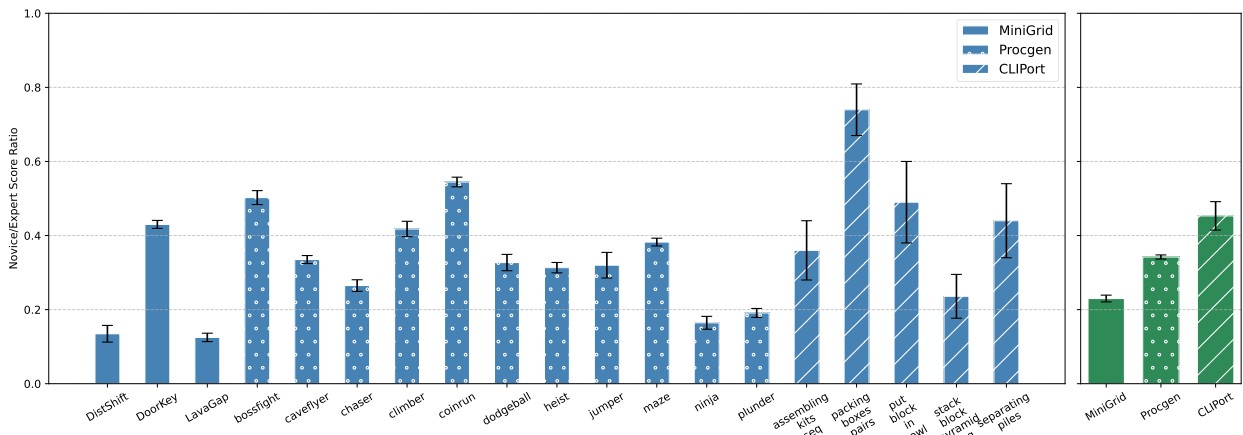

Figure 12: **Left:** Per-environment novice-to-expert score ratios $r_i$ with error bars $2 \times \delta r_i$, hatched by suite (no hatch: MiniGrid; dotted: Procgen; slashed: CLIPort). **Right:** Suite-averaged ratio $r_S$ with aggregated error bars $2 \times \delta r_S$. Novices consistently achieve less than half of expert returns across all domains, and exhibit higher instability (wider error bars) on more stochastic tasks.

Table 2: Wall-clock time on NVIDIA A6000 (48 GB) with 100 GB RAM. Times rounded to nearest hour for readability.

| Category | Training | Evaluation | Total |
|---|---|---|---|
| *By environment* | | | |
| MiniGrid | **3 d**-14 h | **0 d**-12 h | **4 d**-2 h |
| Procgen | **1105 d**-11 h | **14 d**-19 h | **1120 d**-6 h |
| CLIPort | **201 d**-11 h | **21 d**-16 h | **223 d**-2 h |
| *By algorithm* | | | |
| ALWAYS-based | **3 d**-0 h | **6 d**-9 h | **9 d**-9 h |
| THRESHOLD-based | **85 d**-21 h | **12 d**-1 h | **97 d**-22 h |
| OOD-based | **122 d**-16 h | **5 d**-16 h | **128 d**-8 h |
| RLORACLE | **1082 d**-0 h | **5 d**-17 h | **1087 d**-17 h |
| **Overall** | **1310 d**-12 h | **36 d**-23 h | **1347 d**-10 h |

By algorithm, the skyline RLORACLE dominated resource consumption ($\approx 1087$ GPU-days), while our OOD-based method consumed an additional $\approx 128$ GPU-days. The relatively lower cost of the ALWAYS-based and THRESHOLD-based methods ($9 - 97$ GPU-days respectively) highlights that naive baselines are cheaper but far less adaptive.

These figures make clear that replicating and extending our suite of experiments demands significant computational investments, reinforcing the importance of efficient coordination policies that can reduce expert queries without incurring prohibitive compute costs.

### D.7   Analysis of Oracle Proposer Performance in "plunder"

We note the interesting case of the plunder environment, where the oracle proposer slightly underperformed the best heuristic method (Fig. 5). We hypothesize this is due to the high variance inherent in both the procedurally generated environment and the deep RL training process for the RLORACLE proposer. For certain challenging random seeds, the complex oracle proposer may have converged to a slightly suboptimal policy, whereas a simpler heuristic method was more robust.

### D.8 Analysis of the Simulated Validator

We provide additional details and analysis on our simulated validator setup to assess its reliability and sensitivity.

#### D.8.1 Holistic Evidence for Validator Effectiveness

To assess the effectiveness of our simulated validator, one could ideally analyze the direct quantitative correlation between its predicted policy rankings and the true rankings from the oracle validator. However, a more practical and holistic measure of its real-world utility is demonstrated by its impact on final model performance across our entire suite of experiments.

Our primary evidence for the validator's effectiveness is presented in **Finding 2** and detailed in Fig. 4. These results show that methods leveraging our simulated validation approach (denoted by solid bars in Fig. 4) collectively outperform their counterparts in 14 out of 19 environments. Furthermore, 3 of the 4 most successful methods overall are ones that employ the simulated validator. This demonstrates that our simulated validator is an effective and reliable tool for hyperparameter selection in the challenging YRC-0 setting. The consistent performance improvement gained by using it provides strong empirical evidence of its ability to select policies that generalize well to the true test conditions, which is its ultimate purpose.

#### D.8.2 Rationale for the Degree of Novice Weakening

Our simulated validator relies on a "weakened novice" ($\pi_{\tilde{n}}$). Our choice to weaken the novice by training it with 50% of the data/epochs of the full novice ($\pi_n$) was a deliberate methodological decision intended to strike a crucial balance.

A more severely weakened novice (e.g., 25% of original training) might perform so ineffectively that it fails to represent the subtle, near-distribution failures of the true novice, leading the validator to learn an overly pessimistic and simple strategy. Conversely, a less weakened novice (e.g., 75% of original training) would be too competent, providing an insufficient performance gap against the simulated expert (the full novice, $\pi_n$) for the validator to learn a meaningful distinction between reliable and unreliable actions.

Therefore, the 50% mark was chosen to create a challenging yet representative simulation where the weakened novice is prone to failure but still retains enough competence to model realistic decision-making challenges under distribution shift.

