# OpenReview forum: "YRC-Bench: A Benchmark for Learning to Coordinate with Experts"
_TMLR — Accepted by TMLR_

### Review · Reviewer_qeVG · 2025-10-17

**Summary Of Contributions:**

This paper makes three primary contributions to the field of human-AI collaboration and reinforcement learning.

1. **Formal Definition:** It formally defines the *"Yield-or-Request Control under Zero-Shot Generalization (YRC-0)"* problem, where an AI agent must learn when to ask for help from an expert in new environments without any interaction with experts.
2. **Benchmark Contribution:** It introduces **YRC-Bench**, an open-source benchmark designed to facilitate research on the YRC-0 problem, complete with a variety of simulated environments, experts, and evaluation tools.
3. **Empirical Study:** Finally, the paper presents a large-scale empirical study on YRC-Bench, evaluating several baseline methods and offering insights into their performance and limitations.

---

**Key Strengths**
- **Addresses a Critical Problem:** The paper tackles the highly relevant and important challenge of how to build AI agents that can safely and effectively leverage expert assistance in novel situations.
- **Comprehensive Benchmark:** The development of YRC-Bench is a contribution. It provides the research community with a set of environments for studying the YRC-0 problem.
- **Thorough Experimental Evaluation:** The authors have conducted an extensive set of experiments, which demonstrates a serious effort to understand the performance of the baseline methods and the challenges of the YRC-0 problem.


---

**Key Weaknesses**
- **Unrealistic Assumptions:** A major limitation is the benchmark's reliance on *“nearly ideal”* simulated experts. The experts for MiniGrid and Procgen are PPO policies trained directly on the test task distribution, and the CLIPort expert is a *rule-based oracle.*
This design choice means the experts have perfect, oracle-like knowledge of the exact environments the agent is being evaluated on — far from a realistic human expert, who would be suboptimal and biased. This assumption fundamentally limits the paper's claims about real-world human-AI collaboration, as the agent is learning to coordinate with a “god-like” oracle, not a realistic partner.
- **Uninformative Baselines:** The *“Always-Yield”* and *“Always-Request”* baselines are so simple that they are designed to be easily beaten and don’t represent any intelligent strategy. Their inclusion is standard, but they don't provide much insight. The fact that a *RANDOM* policy is *“surprisingly strong”* is a major red flag—it indicates that the other, more complex methods are failing to learn a meaningful policy.

**Audience:**

Yes

**Audience Explanation:**

Researchers in AI safety, robustness, and human-AI Interaction, and the general reinforcement learning and decision-making community

**Claims And Evidence:**

No

**Claims Explanation:**

**Claim: The proposed "proposer-validation" decomposition is a framework for tackling the YRC-0 problem.**
- **Evidence Provided:** The paper introduces a two-part framework where a "proposer" (the agent's policy) suggests an action and a "validator" decides whether to execute that action or query the expert. This is presented as a key methodological contribution.
- **Why the Evidence is Unconvincing and the Framework is Inappropriate:**
- **1.** The paper motivates its work by correctly identifying the challenge of deploying AI in "real-world environments" where they must collaborate with "experts, whether humans or highly capable AI systems." It then implicitly argues that its "proposer-validation" framework is a useful model for understanding and tackling this real-world problem.
However, the evidence for this framework's usefulness is unconvincing because the framework itself is an oversimplification of the problem it claims to address. The framework imposes a rigid, binary decision structure (either accept the agent's action or query the expert). A human expert might not just provide an alternative action; they might offer guidance, adjust the agent's plan, or take over for a sequence of actions. The framework's design is too restrictive to capture these richer forms of interaction.
- **2.** The claim that the "proposer-validation" framework is a key contribution to tackling the YRC-0 problem is weak. The core problem, an agent needing to detect its own incompetence in a novel (OOD) environment and cede control to an expert, is already a central topic of research in fields like: 1) Safe Reinforcement Learning (e.g., identifying unsafe states and reverting to a safe policy); 2) Uncertainty Quantification (e.g., using model uncertainty to trigger a specific action); 3) Out-of-Distribution Detection (e.g., detecting covariate shift and flagging for human review).
Meta-Learning RL (e.g., training agents to learn adaptation strategies that can be applied to new, unseen tasks, which is the very essence of the YRC-0 "zero-shot" challenge)
Researchers in these areas are already developing algorithms (like deep ensembles, Bayesian methods, etc.) that are designed to solve this exact issue, and they can be applied without adopting the "YRC-0" or "proposer-validation" labels.

**Claim: The experiments conducted on YRC-Bench provide a generalizable evaluation of methods for the YRC-0 problem.**
- **Evidence Provided:** The paper presents a large-scale computational study across three distinct environments (MiniGrid, Procgen, CLIPort) with multiple baselines, evaluated on task performance penalized by expert query costs.
- **Why the Evidence is Unconvincing and the Experiments are Inappropriately Designed:**
- **1.** The most significant design flaw is the use of "nearly ideal" simulated experts. This is an explicit choice mentioned in the paper. The difficulty often lies in dealing with an expert who is fallible, inconsistent, has communication overhead, or whose strategy may differ from the agent's. By removing this complexity, the experiments test the agent's ability to recognize its own uncertainty, not its ability to coordinate with a realistic expert. The evidence gathered is only valid for a scenario that does not exist in the real world.
- **2.** The paper reports that a "RANDOM" baseline (choosing randomly whether to yield or not) is "surprisingly strong." A well-designed benchmark should present a challenge where intelligent methods clearly outperform naive or random ones.

**Requested Changes:**

**Would Strength the Work**
Adjustment suggestions:
- **1.** The paper's introduction and abstract explicitly motivate the work by highlighting the need for AI agents to collaborate with "experts, whether humans or highly capable AI systems" in "real-world environments." This frames the paper's goal as understanding and improving this real-world human-AI collaboration. Given this stated goal, the complete absence of human participants is a critical weakness. While a large-scale real-world deployment is not expected, even a small, qualitative study with human participants interacting with an agent in one of the simpler environments (e.g., MiniGrid) would add significant value. This could provide a crucial sanity check on whether the findings from the simulated-expert experiments have any bearing on the complexities of real human interaction.
- **2.** It is critical to include comparisons against more sophisticated methods from related fields, such as: baselines from Bayesian deep learning or ensemble methods that are specifically designed for more robust uncertainty quantification. Baselines from active learning or meta-learning that are designed to adapt quickly to new distributions. Without stronger baselines, the conclusion that the problem is difficult is not fully supported; it may be that the tested methods are simply not the right ones.

---

> ### Author Response · Authors · 2025-11-05
>
> We thank the reviewer for their time and critical engagement with our paper. The review raises several fundamental points about our experimental design and framing. We would like to address these concerns directly, as we believe they stem from a deliberate methodological choice that is central to our paper's contribution.
>
> - **The use of "nearly ideal" simulated experts is unrealistic and undermines the paper's claims about real-world human-AI collaboration**:
> We agree that we overemphasized the human-AI collaboration angle in our original framing and will revise the paper to sharpen our claims. The reviewer is correct: our work studies AI-AI collaboration with perfect experts, not the full complexity of human-AI interaction. Our contribution focuses on a critical prerequisite: AI self-assessment under distribution shift. The goal of our large-scale evaluation across 19 diverse environments is not to claim direct generalizability to all human-AI settings, but to robustly test for consistency and prevent our findings from being an artifact of a single domain. To this end, we will revise our paper to: (1) Reframe our work as addressing "AI self-assessment and risk detection under distribution shift," rather than general "human-AI collaboration"; (2) Position the use of ideal experts as a deliberate methodological choice to address the prohibitive cost and scalability of experiments with human participants, which allows us to isolate the core self-assessment challenge; (3) Explicitly state that extending this work to realistic human experts is an important future direction.
> - **The "proposer-validation" framework is an oversimplification and the core problem is already well-studied in other fields (Safe RL, UQ, OOD Detection)**:
> We appreciate the reviewer highlighting the deep connections to Safe RL, uncertainty quantification, and OOD detection. We agree that we have not adequately positioned our work relative to these established fields. We will make the following revisions: (1) Expand our Related Work (Section 2) to provide a thorough discussion of how YRC-0 relates to these areas, acknowledging the substantial prior work on detecting agent incompetence and uncertainty; (2) Clarify that we do not claim the "proposer-validator" decomposition as a novel algorithmic framework, but rather as a crucial diagnostic tool that our benchmark enables. As our Finding 5 (which after revisions, is now Finding 4) demonstrates, this decomposition allows us to pinpoint whether failures stem from poor proposals or poor validation—an insight that is difficult to obtain without such a standardized setup. Our contribution is the formulation, benchmark, and diagnostic framework that together enable deeper, more comparable research into this critical, cross-cutting problem.
> - **The strong performance of the RANDOM baseline is a "major red flag"**:
> We agree this is a striking result that requires careful interpretation. In the extremely difficult OOD setting of YRC-0, the key finding is not that RANDOM performs well, but rather that sophisticated methods that rely on heuristics learned from the training distribution perform poorly, often worse than random chance. This reveals a fundamental failure: these methods become confidently wrong under distribution shift, making systematically poor decisions about when to request help. RANDOM avoids this systematic bias. The fact that methods designed for uncertainty estimation are outperformed by a non-adaptive baseline is not a flaw in our benchmark; it is strong evidence that existing methods fail catastrophically at the core OOD self-assessment challenge we study. We will revise the discussion of Finding 3 (Section 6.2) to emphasize this interpretation.
>
> Regarding the requested changes for human evaluation and stronger baselines, we agree these are excellent and necessary directions for future work. Our paper aims to establish the foundational problem and a stable, scalable benchmark as a prerequisite. As mentioned, our simulation-based approach was a deliberate choice to overcome the cost and logistical barriers of human-in-the-loop experiments. Now that our work provides a robust testbed, it paves the way for future studies to incorporate human participants or compare against more sophisticated baselines from the fields mentioned. We will explicitly add both of these points to our Conclusion & Limitations section to guide the next wave of research in this area.

---

> > ### Comment · Reviewer_qeVG · 2025-11-05
> >
> > Thanks authors to address my concerns and have agreed to make the right changes. My previous concerns are addressed by the proposed revision plan. Specifically, I am now satisfied with the plan for 1) reframing the paper's main pitch as "AI self-assessment under distribution shift", 2) situating their working with existing fields as a diagnostic tool for the benchmark, 3) interpreting more about RANDOM baseline.

---

> > > ### Author Response · Authors · 2025-11-12
> > >
> > > We thank the reviewer for engaging deeply with our work and for their constructive feedback. Their critical insights were invaluable in helping us sharpen the paper's core claims and better position our contributions. We are pleased that our revision plan has addressed the reviewer's concerns.

---

### Review · Reviewer_8Tux · 2025-10-19

**Summary Of Contributions:**

The primary contribution is the introduction of YRC-0, a new, challenging variant of the YRC problem. In YRC-0, an agent must learn a coordination strategy for new environments/domains in an unsupervised setting i,e. without interacting with the domain expert during the training phase. This setting is motivated by the high cost of expert supervision and the need for robust, generalizable solutions.

To further enable work on YRC-0, the authors also introduce a new open source benchmark called YRC-Bench.

The authors provide a baseline implementation while conducting large scale evaluations to kick start work on YRC-0 and YRC-Bench.

Strengths:
- The problem description is relevant and timely for real world AI deployment. The constraints and framework of YRC-0 make sense. Expert abilities may improve, new experts may come up, while an agent is already deployed and unaware of those experts. Minimizing the cost of calling experts is also a sound real world assumption.
- The proposed benchmark YRC-Bench appears to have all the elements that could drive its adoption, i.e. 3 environments, gym-like API, simulated experts, baseline implementation and an evaluation pipeline.
- The baseline implementation and analysis from the authors seems to be very meticulous, and provides hints for future work, for e.g. further work needed on proposers over validators.
- A bit of a meta point, but one of the strengths of the work is that it itself lists down a lot of limitations, assumptions, and directions for future work :)

Weaknesses
- To be fair, the authors list most of the weaknesses below in Section 7.
- Proposer-validator framing. While this framework does make sense in the paper, the authors mention their bottleneck findings could be useful for future work on this problem involving other methods like offline RL or meta leaning. This point is not clear to me. How would this framing fit with these other methods? Not all potential solutions for YRC-0 can fit this framework. For e.g. either active RL  or meta leaning methods. In meta-learning, there is no clear distinction between proposing and validating.
- Usage of simulated experts and validators. It is not clear how the results on this benchmark will transfer over to real world AI systems with experts.
- Lack of human evaluation. Lack of human evaluation is understandable from the point of view of lack of resources, but any limited human eval would lend more weight to the results in the paper. Especially the results which the authors claim can guide further work in this direction.
- The paper motivates the problem as one of human and AI coordination, but the nature of the coordination is simplified to choosing one of two policies. A more realistic form of coordination could be future work perhaps.

**Audience:**

Yes

**Audience Explanation:**

Yes, it is clear that the YRC-0 problem defined by the authors is challenging, novel, interesting and worth studying. The open-source (to be ?)  benchmark proposed is a great start towards more work in this direction, and hopefully TMLR’s audience can contribute towards better candidate submission and maybe removing some of the assumptions in the benchmark too.

The ablations in the paper, even though limited by some assumptions, are very thorough and can provide ideas about future work.

**Broader Impact Concerns:**

Research into AI and human collaboration is very important, and the authors attempt to advance this direction is commendable. In the light of recent advancements in AI and dangers of potential AI job automation, more work on AI and human collaboration is needed!

**Claims And Evidence:**

Yes

**Claims Explanation:**

Most of the claims made by the paper seem to be convincing. Here, I list some of the claims that need a bit more clarity:
- The authors claim that YRC-Benchmark is open-source, but I don’t see any mention of the link to the open source implementation. Perhaps the authors are waiting to do so after submission, in order to support the anonymization of the review?
- The claim made in the paper about proposer-validator bottlenecks being useful for future research of methods like meta-learning, offline RL. This might be true only for the narrow set of proposers and validators considered by this study. The proposers considered by this paper are fairly simple heuristics, and it is completely plausible that different / stronger proposers would reveal new challenges for validators.
- Going on about the same claim above, looking at Figure 5, it seems to be hard to draw the conclusion about validators not being a bottleneck. A lot of the error bars overlap, so it’s hard to draw any conclusion either way. But yes, the point about replacement oracle validators being more impactful, is completely valid.
- The claim that simulated validators are an effective way to tune hyperparameters. The authors further claim that replacing simulated validators with an oracle doesn’t change results significantly. While the simulated validator setup is clever, it would be informative to know more details here. Do the simulated validators correlate with true test performance? How does the setup depend on how much the novice is weakened? Details about how the novice is weakened would be useful to know.

**Requested Changes:**

More details about the simulated validator setup would be great to have in the paper.
- Some analysis around, how well the simulated validator correlates with true test time outcome.
- Does the magnitude of the weakness of the simulated novice impact the effectiveness of the simulated validator.

Tighten the proposer-validation bottleneck claim. Looking at figure 5, why do some environments cause test performance to actually go down after using oracle proposer, for e.g. plunder? A deep dive into that might be interesting.

---

> ### Author Response · Authors · 2025-11-05
>
> We sincerely thank the reviewer for the positive and highly constructive feedback on our work. The reviewer’s suggestions are invaluable for strengthening the paper, and we plan to incorporate them as follows.
>
> - **Open source implementation of YRC-Bench**:
> As the reviewer mentioned, we planned to open-source the implementation once the decisions are out. However, to show our commitment and transparency, we have added the URL to the anonymized code in the revision.
> - **The proposer-validator bottleneck claim is not clearly supported by Figure 5 due to overlapping error bars**:
> We agree completely with this assessment. Our intention was to highlight the relative difference in performance gains, and we concede that our original claim of validators "not being a bottleneck" was too strong and not fully supported by the statistical evidence. The key finding, which remains valid, is that improving the proposer offers a much larger potential for performance gain across most environments. The reviewer also raises an excellent question about why performance sometimes decreases with an oracle proposer (e.g., in "plunder"). We hypothesize this is due to the high variance inherent in both the plunder environment and the deep RL training process for the RLORACLE itself. For certain difficult seeds, the oracle proposer may have converged to a slightly suboptimal policy compared to the best policy found by the simpler heuristic methods. We will revise the text in Finding 5 (which after revisions, is now Finding 4) to be more precise. We will rephrase our claim to state that our results suggest improving the policy proposer offers a much larger potential for performance gain than improving the validator. We will also add a brief note in the appendix discussing the plunder anomaly, crediting the review for prompting this deeper dive.
> - **It is not clear how the proposer-validator framing would fit with other methods like offline RL or meta-learning**:
> This is a crucial point that deserves clarification. We did not intend for the proposer-validator decomposition to be a rigid algorithmic requirement that all future methods must adopt. Rather, we introduce it as a general lens for understanding a wide range of methods. We included meta-learning in our study to be comprehensive, as it is a strong, general framework applicable to many problems. Fundamentally, any framework that leverages optimization can be viewed through this lens: the optimization process itself proposes a set of parameters (or policies) to minimize a loss function, which serves as the validator.
> - **More details about the simulated validator setup are needed**:
> We thank the reviewer for this excellent suggestion. Upon revisiting this point, we realized the most compelling evidence for our validator's effectiveness was already present in our main results. We have therefore added a new, detailed subsection to the appendix (App. D.8) to address the reviewer's questions directly.

---

> > ### Comment · Reviewer_8Tux · 2025-11-05
> >
> > Thank you to the authors for addressing the concerns in my review. The changes look good to me! To be more specific, I like the caveats added around the discussion of proper-validator framework, and I still believe the main strength of the paper is in the open source benchmark, which can hopefully help the development of more varied, perhaps stronger benchmarks!

---

> > > ### Author Response · Authors · 2025-11-12
> > >
> > > We thank the reviewer for their positive feedback. We are glad our revisions addressed the concerns and appreciate the constructive review.

---

### Review · Reviewer_VUBK · 2025-10-28

**Summary Of Contributions:**

This paper considers an MDP setting where a "novice" policy can perform well on tasks that were in the training distribution, but performs poorly on OOD tasks. However, there is an "expert" policy which performs well on OOD tasks. The goal of the paper is to learn a coordination policy, which will decide when to use the novice policy and when to defer to the expert policy (trying to minimize the use of the expert, as this is assumed to be costly). A key distinction from prior related work (on the YRC problem) is that in this paper, at train time there is no expert available (YRC-0).

In particular, the authors model the cost of using the expert as effectively a fraction of the expert's return (i.e. as they say, one might treat rewards gained under expert behavior as worth 50% of their actual reward). This is a reasonable, but very specific, way to model the issue of cost.

The authors present a new benchmark, YRC-Bench, which adds expert policies to a bunch of reasonable baseline MDPs. To model "perfect" performance, they can't exactly solve the problem, but can run costly RL training to get something like an optimal policy.

The authors frame their problem in terms of choosing a policy proposer (which creates a set of policies) and a policy validator (which picks one from the set). Their observation is that getting a policy validator is much more difficult given the lack of access to experts at train time. A clever but slightly ad-hoc solution to this is to treat the "novice" policy as an expert on the training tasks, create a deliberately degraded policy, and then train the coordinator to coordinate between those, hoping that OOD it will still generalize between the true expert and the true novice.

Experimental results are positive in the sense that the benchmark and methodology seems to be reliable, but algorithmically there is much room for improvement (which is not necessarily a bad thing in a paper that mainly wants to present a new task and benchmark).

**Audience:**

Yes

**Audience Explanation:**

The YRC task, and this variation of it, and related work in imitation learning etc., are of longstanding interest to the ML community; people will likely find the new benchmark useful.

**Broader Impact Concerns:**

There are no broader impact concerns; this is a paper fairly detached from real-world deployment.

**Claims And Evidence:**

Yes

**Claims Explanation:**

The problem setting is reasonable and connected well to previous work. Everything is explained clearly. Experiments are solid and support the claims in the paper. The authors are very honest about their limitations.

**Requested Changes:**

The paper "Fully General Online Imitation Learning" (Cohen et al., JMLR 2022) takes a very different perspective but deals with a related problem setting. The authors may want to (or may reasonably decide not to) cite it.

---

> ### Author Response · Authors · 2025-11-05
>
> We sincerely thank the reviewer for their review, and appreciate that the reviewer found our problem setting reasonable, our experiments solid, and our claims well-supported. We have addressed the reviewer's suggestion below.
>
> - **The paper "Fully General Online Imitation Learning" (Cohen et al., JMLR 2022) may be a relevant citation**:
> We have reviewed the paper and agree that it is highly relevant to the broader landscape of learning with expert guidance. While its theoretical perspective is indeed different (focusing on online learning and regret bounds, where an expert is available for queries during the learning process), it addresses the core theme of deciding when to rely on an expert. Situating our work in contrast to this online setting helps to further clarify the unique challenges of our zero-shot, offline-training formulation (YRC-0). We will add a citation and a brief discussion of Cohen et al. (2022) to our Related Work section (Section 2).
> - **On the cost model being reasonable, but very specific**:
> We agree with this characterization. Our choice of modeling the expert cost as a fraction of the expert's return was a deliberate design decision for the benchmark. We aimed for a formulation that was both interpretable and automatically adaptable across environments with vastly different reward scales. We will ensure our text in Section 4.2 reinforces that this is one of several possible cost formulations, chosen for these practical benefits in a benchmark setting.
> - **On our simulated validator being clever but slightly ad-hoc**:
> This description perfectly captures the core difficulty of the YRC-0 problem: without access to the true expert or test distribution, any validation method must, by necessity, rely on some form of heuristic simulation. Our approach is a pragmatic first attempt, and its effectiveness was one of the positive findings of our study. We fully agree that developing more principled and less ad-hoc validation strategies is a key direction for future work, which our paper and benchmark now enable.
>
> Finally, we are very encouraged that the reviewer find our benchmark and methodology to be reliable and that the reviewer believe the community will find YRC-Bench useful. Fostering future research in this area was our primary goal.

---

### Author Response · Authors · 2025-11-05

We sincerely thank all reviewers for their thoughtful and constructive feedback. We have carefully revised the paper in response to each comment and implemented the suggested improvements throughout the manuscript.

All revisions and additions are highlighted in **green and underlined** in the new version to make it easier to identify the changes. These include clarifications to our claims, expanded discussions in the Related Work and Limitations sections, improved explanations of the proposer-validator framing and simulated validator setup, and the inclusion of new citations and detailed appendices.

We deeply appreciate the reviewers' time and insights, which have significantly strengthened the paper.

---

### Decision · Action_Editor_og3C · 2025-12-14

**Recommendation:** Accept with minor revision

**Additional Comments:**

The reviewers generally liked the work, and the authors addressed some concerns. Given this, the decision is Accept with Minor Revisions. However, there are still several issues with the work in its current form, that should be addressed before the paper can be fully accepted. I can check these issues, without needing to resend for review.

First, the problem setting needs to be more clearly identified upfront. The structure seems to be: train a novice policy in a set of training tasks (where it is actually an expert), and then assume it is deployed to test tasks where it performs poorly. Why does it have to perform poorly on the test tasks? Can’t we just assume it may not be as effective, as these tasks could be OOD? Additionally, if we know the novice policy may not be effective in the OOD tasks, why restrict it from continuing to learn on the test tasks?  Especially, we could update with good actions given by the expert. Similarly, the coordination policy could also update with new information, including about the expert. There may be a good reason these cannot be updated. It might even be possible that this is not mimicking a real scenario, and the goal is just to understand the quality of the coordinator learned after training, by testing it fixed in a test setting. But, again, these choices/restrictions were not clearly motivated. I understand that in zero-shot RL, it is common to deploy a fixed policy. But, in this new scenario proposed in this work, it is good to motivate and clearly outline what this specific problem setting with its restrictions is mimicking, with examples of real world scenarios.

Second, the choice of the methods in the comparison is not fully explained. A reviewer rightly points out the connections to other fields, that effectively use uncertainty estimates to guide decisions. There are many potential approaches that could be used, though they might need to be adapted to this setting. The one that seems to have been chosen is deep SVDD. Why deep SVDD? The approach is not described nor is it motivated why this is chosen as one of the more complex approaches.

Third, the authors now make a claim that the separation into proposer-validator gives a clear diagnostic approach, but this reasoning remains unclear to me. A different class of algorithms may not use this separation, so it seems like the analysis just provides some insight into the specific choices you made here in this proposed-validator class of algorithms. I think it could be ok to largely restrict focus to this class of algorithms (proposer-validator), though this should also be justified. But it does not seem quite right to say that you make this choice for diagnostic reasons. There is a claim that all (or most) methods can be cast this way, such as gradient-based methods with loss functions. If this is the case, you could explain why this way of looking at it helps diagnose such a gradient-based approach in general.

Finally, the way the benchmark is set-up, the simulated novice uses the same training tasks as the true novice, but simply with less supervision. The tasks themselves will not be OOD. Why not use a subset of the training task and then use the rest as test tasks? I understand that the approach is just one proposal, and the goal of the work is not necessarily to propose an optimal algorithm for your benchmark. Nonetheless, under the restrictions you’ve place for your problem setting, you should motivate why this proposed algorithm makes sense.

Note: the title is Learning to Coordinate with Experts
which is quite broad. But I believe the primary contribution here is the benchmark. The title should reflect this.

**Audience:**

Yes

**Audience Explanation:**

The problem setting of coordinating with an expert is useful and this paper develops a benchmark for a specific scenario. Creating sensible and useful benchmarks is difficult, and important, and this paper was systematic in the benchmark choices.

**Claims And Evidence:**

No

**Claims Explanation:**

There are some points that need clarification (see additional comments).

---

> ### Author Response · Authors · 2025-12-24
>
> We thank the Action Editor for the constructive feedback and the decision to accept with minor revisions. We have updated the manuscript to address all raised points. Below is a summary of the changes:
>
> **1. Motivation for the Problem Setting (No Online Learning)**
> *   **AE Comment:** Requested clearer motivation for why the agent cannot update parameters during testing, with real-world examples.
> *   **Response:** We have updated **Section 3** to explicitly motivate the "frozen policy" constraint. We clarify that in high-stakes applications (e.g., medical diagnosis, autonomous driving), online trial-and-error is often prohibited due to immediate safety risks. Furthermore, we note that regulatory frameworks often require pre-deployment certification, making dynamic parameter updates legally problematic. We also clarified that the "poor performance" on test tasks is not a contrived requirement, but the natural consequence of significant distribution shift, which is exactly when expert assistance becomes necessary.
>
> **2. Justification for Deep SVDD**
> *   **AE Comment:** Asked for the rationale behind selecting Deep SVDD.
> *   **Response:** We have added text in **Section 5** motivating this choice. We selected Deep SVDD as a representative approach because it learns a compact decision boundary in the latent space without requiring computationally expensive generative modeling of high-dimensional observations (like pixels). This makes it practically suitable for post-hoc application to the internal representations of pre-trained RL agents, compared to ensemble or Bayesian methods which often require architectural changes or multiple training runs.
>
> **3. Proposer-Validator Decomposition**
> *   **AE Comment:** Questioned the generality of the proposer-validator claim and suggested it is better framed as a specific choice for this analysis.
> *   **Response:** We have refined the text in **Section 3** to clarify the scope of this claim. As suggested, we now introduce this decomposition primarily as a **diagnostic framework** for the class of candidate-selection methods analyzed in this work. This framing allows us to empirically isolate whether a method's failure stems from the inability to *generate* good policies (proposer failure) or the inability to *select* the best one (validator failure), without claiming it as a rigid requirement for all future algorithmic classes.
>
> **4. Simulated Novice Construction**
> *   **AE Comment:** Asked why we use limited supervision rather than holding out training tasks to simulate distribution shift.
> *   **Response:** We have clarified this design choice in **Section 5**. While holding out training tasks is a valid alternative, we prioritized allowing the primary novice to learn from the full breadth of available training data ($\mathcal{E}_{\text{train}}$) to maximize its baseline capabilities. We therefore use "limited supervision" (under-training) as a heuristic proxy to simulate the performance gap caused by distribution shift, allowing us to approximate the drop in competence without artificially handicapping the primary agent's data access.
>
> **5. Title Change**
> *   **AE Comment:** Suggested the title should reflect that the primary contribution is the benchmark.
> *   **Response:** We agree and have changed the title to: **"YRC-Bench: A Benchmark for Learning to Coordinate with Experts"**.
>
> We believe these revisions fully address the remaining concerns and have uploaded the de-anonymized camera-ready version.